# Immediate Effects of Foam Roller and Stretching to the Lead Hip on Golfers Swing: A Randomized Crossover Trial

**DOI:** 10.3390/healthcare11142001

**Published:** 2023-07-11

**Authors:** Yuji Hamada, Kiyokazu Akasaka, Takahiro Otsudo, Yutaka Sawada, Hiroshi Hattori, Yuki Hasebe, Yuto Kikuchi, Toby Hall

**Affiliations:** 1Graduate School of Medicine, Saitama Medical University, 981 Kawakado, Moroyama 350-0496, Japan; y.hamada@ymail.ne.jp (Y.H.);; 2Department of Rehabilitation, Kawagoe Clinic, Saitama Medical University, 7-21 Wakitahontyo, Kawagoe 350-1123, Japan; 3School of Physical Therapy, Saitama Medical University, 981 Kawakado, Moroyama 350-0496, Japan; 4Department of Physical Therapy, School of Health Sciences, Tokyo University of Technology, 5-23-22 Nishikamata, Tokyo 144-8535, Japan; 5Department of Rehabilitation, Saitama Medical Center, Saitama Medical University, 1981 Kamoda, Kawagoe 350-8550, Japan; hasebe_y@saitama-med.ac.jp; 6Curtin School of Allied Health, Curtin University, Kent Street, Bentley, WA 6102, Australia

**Keywords:** lumbar spine, hip joint, golf, three-dimensional analysis

## Abstract

Golfers with decreased range of motion (ROM) of their leading hip internal rotation (IR) have increased lumbar rotation ROM and load. This study investigated the effects of foam roller (FR) applied to their leading hip muscles combined with stretching to the leading hip together with lumbar rotation ROM during the golf swing. The study design was a crossover design. Subjects were allocated to one of two groups comprising FR and dynamic stretching (FR + DS) or practice swing. Motion analysis was used to evaluate hip and lumbar angles during the golf swing. Data were compared using analysis of variance with Bonferroni correction using paired *t*-test’s post hoc. The association between lead hip IR angle and lumbar spine left rotation (Lrot) angle was investigated using correlation analysis. Lead hip IR ROM during the golf swing was significantly greater in the FR + DS group (*p* = 0.034). The FR + DS group showed a moderate negative correlation between lead hip IR ROM and lower lumbar spine Lrot ROM during the golf swing (*r* = −0.522). The application of FR + DS might be useful to increase lead hip IR angle during the golf swing. Moreover, the application of FR + DS improves lead hip IR angle and may decrease lumbar spine rotation.

## 1. Introduction

Approximately 80% of golf injuries can be related to overuse, with low back pain (LBP) being the third most common injury among amateur players [1]. It is suggested that increased practice time and possibly carrying a golf bag during a round of golf are factors that may increase lumbar injuries in golfers.

Sahrmann et al. proposed the concept of movement system impairment syndromes [2]. This concept purports that bodily alignment and repetitive motion cause musculoskeletal injury, ultimately leading to pathologic anatomical changes in tissues and joint structures. To our knowledge, repeated stress in the spinal column causes micro-failure of surrounding tissue, which may be cumulative and may lead to fatigue failure [3]. Compared to the thoracic spine, the lumbar spine has less available rotational range of motion (ROM) due to anatomical differences. In the lumbar spine, facet joint damage can occur when there are more than three degrees of rotation per vertebral segment [4]. Golf is a repetitive activity in which the same motion of swinging is repeated. The motion of the golf swing is primarily the rotation of the spinal column, which is likely to cause a high degree of stress on the lumbar spine. A previous study measured spinal loading during the golf swing and showed that shear loading is 329.36 N ± 141.27 for professional golfers and 596.14 N ± 514.01 for amateur golfers [5]. Amateur golfers, therefore, exhibit a higher degree of spinal loading compared to professional players. Furthermore, golfers with LBP reportedly have more trunk rotation during the golf swing [6]. Hence, it is optimal for golfers to reduce lumbar spine rotation in order to prevent lumbar injuries from rotation overstress. A particular challenge appears to be reducing lumbar rotation during the golf swing in amateur golfers.

One method suggested to reduce the burden of the golf swing on golfers’ lumbar spine is lumbar support. Hashimoto et al. reported that wearing lumbar support may decrease lumbar rotation ROM and reduce loading [7]. Another method is lumbar stabilization exercises such as side bridges, bird dog exercises, hip internal rotation mobilizations for 3 sets of 10 repetitions, or hip trigger point releases for 30 s [8,9]. However, the effect of exercise on lumbar spine motion during the golf swing has not been clarified. During the golf swing, golfers with decreased leading hip internal rotation (IR) ROM have been shown to have significantly increased lumbar rotation ROM [10]. Furthermore, both professional and amateur golfers with a history of LBP show a decrease in leading hip IR ROM [11,12]. This suggests that decreasing lead hip IR ROM may contribute to increasing the burden on the lumbar spine in golfers. LBP and limited hip rotation ROM has also been shown to be associated with rotation sports [13]. Thus, clarifying the effect on the lower back of improving hip ROM may be of value.

Stretching has been reported in many studies to increase ROM, prevent injury, and improve sports performance [14,15]. Stretching is a frequently performed method of exercise therapy advocated by physical therapists as well as part of warming up prior to sports participation. Dynamic stretching (DS) is favored in golf because it has a better effect on performance than static stretching [16]. In recent years, self-soft tissue massage has become a common intervention used in physical therapy and rehabilitation of musculoskeletal disorders purported to increase myofascial mobility [17]. Self-massage tools include the foam roller (FR). Users apply pressure to the soft tissues of the target area with their body weight while lying on the FR and performing a rolling motion. Through this process, the soft tissues are stretched by both direct pressure and friction. Physiological mechanisms underlying FR include an increase in soft tissue blood flow as well as a decrease in tissue hardness and muscle viscoelasticity [18,19,20]. These effects are considered to be useful in increasing joint ROM, improving performance, and reducing delayed onset muscle soreness [21,22,23,24]. Therefore, FR is increasingly being used by athletes when warming up, cooling down, and rehabilitating. The combined used of FR and dynamic stretching exercises has been shown to be useful for increasing flexibility [25]. Stretching the hip muscles has also been reported to increase hip mobility as well as decrease lumbar spine mobility during exercise [26,27]. This suggests that targeting hip mobility may reduce the loads on the lumbar spine. However, these previous studies have focused only on changes in hip flexion and extension mobility. Consequently, the changes in hip rotational ROM that occur when hip joint mobility is increased, and the effect that this has on the lumbar spine are unknown. The effects on hip and lumbar spine motion during the golf swing motion are also not clear.

Clarification of the impact of increasing leading hip ROM on lumbar spine motion during the golf swing may provide preliminary evidence for a method of reducing LBP. The purpose of this study was to investigate the effects of leading hip FR combined with DS on kinematics during the golf swing. We hypothesized that increasing the leading hip IR ROM by FR combined with DS would decrease lumbar rotation ROM during the golf swing.

## 2. Materials and Methods

### 2.1. Study Design and Ethical Considerations

The study was a randomized crossover design (Figure 1). Participants were divided randomly into one of two groups. Group A practiced their swing (SW) in Period I and undertook FR and DS (FR + DS) in Period II. Group B followed the opposite procedure to Group A. The primary and secondary outcomes were evaluated pre- and post-intervention in Periods 1 and II. Measurements at each evaluation session took approximately one hour. The participants were instructed to perform a three-minute pre-golf warm up prior to the measurements. Both groups had a washout period to exclude any intervention effects. This washout was a 1-week gap between the end of Period I and the start of Period II [21]. In Period II, the exercise was opposite to that of Period I, and measurements were taken using the same method as that in Period I. All data were collected by one researcher (YH with more than 7 years of clinical experience). Due to the study design, participants and the researchers who administered and performed the statistical analyses were not able to be blinded. Measurement procedures were performed in the following order: secondary outcome variables, primary outcome variables, intervention, primary outcome variables, and secondary outcome variables. Secondary outcome variables were collected using a computer-generated random number table to avoid bias due to the evaluation order. This study was registered in the UMIN Clinical Trials Registry (UMIN000043159) according to the Declaration of Helsinki and approved by the Ethics Committee of Saitama Medical School (permit number 954).

### 2.2. Participants

Participants were recruited through poster displays at four golf driving ranges between May and November 2021 and were required to provide informed consent prior to measurement and participation in the study. Inclusion criteria were no current episode of LBP and no orthopedic disease under treatment. Subjects were required to be participating in at least five rounds of golf per week, with a best score of 110 or less for 18 holes. Exclusion criteria were those with hip or spinal motor impairment and those who would not give their consent to participate in this study. The average score of the previous five rounds was used as the competition level [7].

### 2.3. Motion Analysis as Primary Outcome

A three-dimensional motion analyzer (Vicon MX System; Vicon Motion Systems Ltd., Oxford, UK) was used, and up to 10 cameras were installed (Figure 2). Maximum angles of the lumbar spine and hip joint that occurred during the golf swing were determined. A thirty-five marker reference set (left and right frontal and back head, seventh cervical, ten thoracics, the jugular notch where the clavicle meets the sternum, xiphoid process, right spine of the scapula, right and left acromion, left and right humerus, left and right forearm, left and right lateral epicondyle of humerus, right and left the stromal process of the radius, left and right stromal process of the ulnar, left and right dorsal second metacarpal, left and right anterior superior iliac spines and posterior superior iliac spines, left and right thighs, left and right lateral epicondyles of the femur, left and right lateral lower thighs, left and right lateral malleolus, left and right dorsal second metatarsals, left and right calcaneal tuberosity) Plug-in-Gait full-body Ai model for the calculation of kinematic data of the lower limb was used [28]. Additionally, for the measurements of spinal motion, a set of ten markers (third thoracic spinous process, left and right seventh thoracic transverse processes, twelve thoracic spinous processes, left and right one lumbar transverse processes, third lumbar spinous process, left and right four lumbar transverse processes, five lumbar spinous processes) were affixed to the spine according to a previous report [29]. Upper and lower lumbar spine motion data were calculated using Visual 3D (C-Motion, Germantown, MD, USA). The sampling frequency was 240 Hz, and the cutoff frequency was 6 Hz [28,30]. The filtering method was a zero-lag fourth-order low-pass Butterworth filter on the raw 3D coordinate data [30]. Participants wore fitted black shorts for data collection. The gear used was their own clubs, shoes, and gloves. On the day of the measurements, subjects were instructed not to practice in advance. Three minutes of warm-up was conducted with no restrictions on intensity or method [31]. The driver shot was performed five times. The tees were of any height, and the players were asked to take full swings. The net was placed approximately 3 m ahead of the golf ball (Titleist Pro V1^®^; Acushnet Holdings Corp, Fairhaven, MA, USA). The subject was instructed to imagine that they were at the golf tee and to hit the ball as straight as possible with the timing of their choice. The subject was allowed to make up to one bare swing per swing. For swing analysis, the end of the swing was defined as the position where the left shoulder joint stopped moving [7]. In general, data from the latter three of the five measurements were used. If the data accuracy was low, the first or second data was adopted, and the data was calculated by averaging the three data points [32]. For right-handed golfers, their left side was Lead and their right Trail. Lead hip maximum IR angle and upper and lower lumbar spine maximum left rotation (Lrot) angle were calculated.

### 2.4. Clinical Test as Secondary Outcomes

An inclinometer was used to measure hip IR and external rotation (ER) ROM (BROM^®^; Performance Attainment Associates, Lindstrom, MN, USA). Movement was evaluated with the participant prone, and both active and passive ROM was evaluated. The starting position was prone, with their hip joint in 0° of abduction and knee joint in 90° of flexion. The inclinometer was placed immediately proximal to the medial malleolus. This method has been shown to have high intra- and inter-tester reliability [11]. The posterior inferior iliac spines were fixed to the bed with a belt, and measurements were taken to avoid pelvic compensations. Three trials were performed, and the average value of the three tests was used.

### 2.5. Intervention Program

FR + DS was provided to the lead hip side. For FR (GRID1.0; Trigger Point Performance Inc., Durham, NC, USA), the starting posture was long sitting, and the target areas were the posterior thighs and buttocks (Figure 3a,b). Based on previous reports, each target site was divided into distal and proximal areas, and each was performed for 1 min [21], for a total of 4 min. The speed was approximately 4 s per round trip, and the intensity was within the range that did not cause discomfort [33].

FR was implemented, followed by DS (Figure 3c–e). DS was performed in the order c, d, and e. Each stretching exercise was based on previous reports [34,35]. First, C’s starting position was with the leading knee on the chair and the leading foot secured by the back of the chair. From the starting position, the lower limb opposite the leading side was raised, the left rotation of the trunk was performed to the end range, and leading hip IR exercises were performed (Figure 3c). Secondly, D’s starting position was with the feet positioned shoulder-width apart, and the golf club grasped with the upper extremity opposite to the leading side. As in C, the left rotation of the trunk was performed to the end range, and leading hip IR exercises were performed(Figure 3d). Finally, E’s starting position was as in D, with the club held parallel to the floor at the pelvis. Keep the leading foot always on the floor, left rotation of the trunk was performed to the end range, and leading hip IR exercises were performed(Figure 3e). Two sets of 15 repetitions (without holding) per set were conducted. The intensity was kept within the range that would not cause discomfort, and the rest period after each set was 15 s [36].

SW practice was based on prior studies [37]. Ten bare half swings of the sand wedge were performed, followed by three full swings with the sand wedge, 8-iron, 5-iron, longest iron or fairway wood or hybrid, and driver. The longest iron or fairway wood or hybrid was chosen by the participant.

### 2.6. Sample Size

To determine an adequate sample size for achieving statistical significance with 80% power (1 − b), we performed power analysis a priori (G*Power 3.1.9.7, http://www.gpower.hhu.de/, accessed on 1 March 2021). The effect size was set to 0.25 (a = 0.05; 1 − b = 0.8) in the F test. As a result of the power analysis, a total of 36 participants were required as spread over blocks (FR + DS pre, FR + DS post, SW pre, SW post). Therefore, we recruited at least 22 participants per group, requiring a total of 44 participants.

### 2.7. Data Processing

Statistical analysis was performed using IBM SPSS statistics for Windows, Version 28.0 (Armonk, NY, USA; IBM Corp Released 2020). Normality was assessed and found to be normally distributed according to the Shapiro–Wilk test. To understand the effects of FR + DS and swing practice, data from subjects who performed FR + DS in Period I and Period II were defined as FR + DS group, and the data from subjects who performed swing practice in Period I and II were defined as SW group. A 2 (exercise: FR + DS vs. SW) × 2 (time: pre vs. post) repeated measures analysis of variance was conducted to test the effects of different conditions on dependent variables. If a main effect or interaction was identified, follow-up analyses were conducted using Bonferroni correction with a paired *t*-test to determine the effect. The amount of change was evaluated with either a paired *t*-test or a Wilcoxon rank test. Spearman’s rank correlation coefficient was used to evaluate the correlation between the amount of change in lead hip IR angle and the lumbar spine Lrot angle during the golf swing pre and post-FR + DS intervention. The significance level for this study was set at 0.05.

## 3. Results

Subjects comprised 22 healthy adult males (mean ± SD: age was 32.6 ± 8.5 years, years of experience was 9.5 ± 6.3 years, BMI 23.0 ± 6.4 kg/m^2^, the five golf rounds average score 97.7 ± 14.3, golf training frequency per year 83.5 ± 94.3 times). The experiment was conducted without side effects, and no participants were excluded. All participants were right-handed, with the left hip on the Lead side. Lead hip IR angle and upper and lower lumbar spine Lrot angle during the golf swing occurred between the follow-through and the end range in all cases.

### 3.1. Differences in Hip and Lumbar Spine ROM before and after Intervention

Lead hip IR angle and lumbar Lrot angle during the golf swing before and after intervention (Figure 4) were compared within and between groups. Lead hip IR angle during the golf swing showed an interaction effect and was significantly greater in the FR + DS group only (FR + DS; pre, 15.7 ± 6.59, post, 16.9 ± 6.88, SW; pre, 15.9 ± 4.53, post, 15.1 ± 5.61, degree, *p* = 0.034, ES = 0.59). However, there was no significant difference in lumbar spine Lrot angle during the golf swing no interaction between the two groups (Upper lumbar Lrot: FR + DS; pre, 7.2 ± 2.41, post, 7.0 ± 2.37, SW; pre, 6.7 ± 2.20, post, 6.6 ± 2.70, Lower lumbar Lrot: FR + DS; pre, 7.1 ± 3.80, post, 7.2 ± 3.69, SW; pre, 6.9 ± 2.30, post, 7.3 ± 3.42, degree, *p* > 0.05). Post-intervention comparison between the groups showed that the FR + DS group had a significantly greater change in lead hip IR angle during the golf swing (FR + DS; 1.2 ± 2.03, SW; −0.7 ± 2.69, *p* = 0.008, ES = 0.62). For the secondary outcomes (Table 1), only the FR + DS group showed significantly greater values in lead hip active and passive IR ROM in comparison with the SW group (*p* < 0.000).

### 3.2. Correlation between Lead Hip IR Angle and Lower Lumbar Lrot Angle Changes during the Golf Swing

The correlation between lead hip IR angle and lower lumbar Lrot angle to the amount of change during the golf swing in the FR + DS group is shown in Figure 5. FR + DS group showed a moderate negative correlation between lead hip IR angle and lower lumbar spine Lrot angle during the golf swing (*r* = −0.522; *p* = 0.013).

## 4. Discussion

This present study aimed to investigate the effects of an intervention directed at increasing leading hip flexibility on the hip joint and lumbar spine angle during the golf swing. We hypothesized that the intervention would increase the IR angle of the lead hip during the golf swing and decrease the lumbar rotation angle.

Lead hip IR angle during golf club swinging was significantly greater in the FR + DS group than that before the intervention. Moreover, the lead hip IR angle was significantly greater in the FR + DS group in comparison with the group that performed SW. Several previous reports have indicated the effects of stretching on hip ROM. These included an increase in hip extension angle during walking and an increase in hip flexion angle during the kicking motion in soccer [38,39]. Therefore, improved hip ROM following stretching may generalize to daily activities, including sports. The results of this study also support these previous findings, although there are differences in the direction of hip movement and sports competition. FR and stretching have been shown to decrease muscle and fascial stiffness [20,40]. The group with decreased passive stiffness in the hip external rotation had a significant increase in hip internal rotation angle during movement [41]. We hypothesized that performing FR + DS reduced tissue stretch and passive stiffness that inhibited hip IR ROM. This may have resulted in an increase in passive and active hip IR ROM. FR has been shown to increase muscle blood flow at the performed area [19], and combined FR and DS can improve agility compared to DS alone [42]. It was suggested that the high-speed rotational motion of the golf swing may have contributed to the increase in lead hip IR angle of the golf club swinging by the combination of FR + DS. Therefore, we suggest that FR + DS to the lead hip increased hip IR ROM contributing to the increase in lead hip IR angle during golf club swinging. Professional and amateur golfers with a history of LBP have been shown to have a decreased lead hip IR ROM [11,12]. Regarding the significance of the results, for golfers with limited ROM of the lead hip, FR + DS may have an immediate effect of increasing hip IR ROM which may help prevent the development of LBP. Dillen et al. showed that athletes with LBP who play rotational sports other than golf have a significantly decreased total hip rotation ROM and left hip rotation ROM [13]. Although we chose to adapt the stretching method to golf in this case, it is suggested that similar trends could be achieved by implementing stretches to the hip joints that are tailored to the characteristics of rotational sports.

There were no significant differences in lumbar spine Lrot angle during the golf swing in the FR + DS and SW groups pre- and post-exercise and between the FR + DS and SW groups. To our knowledge, hamstring stretching increases hip range of motion and decreases lumbar spine angle in heavy lifting and standing trunk forward bending movements [26,27]. However, no significant difference in the lumbar spine angle in running motion was observed after stretching hip flexor muscle groups [43], and no consistent view has been obtained regarding the effect of hip stretching on the lumbar spine. In the present study, the reasons why FR + DS to the lead hip did not significantly affect the lumbar spine could be as follows. Younger golfers have been shown to have a higher standard deviation of peak hip torque values and more variability in swinging motion [44]. It is possible that the variation in golf swing motion between individuals did not significantly affect the lumbar spine, which was not the focus of the exercise. Secondly, the FR + DS was performed on the hamstrings, which may have affected the knee rather than the hip.

Despite these findings, there was a moderate negative correlation between the amount of change in lead hip IR angle and lower lumbar spine Lrot angle during the golf swing pre and post-exercise in the FR + DS group. It has been shown that when lumbar rotation is restricted by wearing a corset, the lead hip rotation angle increases [7]. Additionally, the limitation of hip IR ROM increases the lumbar rotation angle during the gold swing [10]. The golf club swing may rely on a complementary relationship between the hip and lumbar spine to generate sufficient rotation. This fits with our finding that the greater the improvement in hip IR angle during the golf swing after FR + DS, the less the lumbar rotation angle. The downswing phase of a golf swing motion was compared with the lumbar spine, trail hip, and lead hip. The lead hip contributed the most and, after the downswing, was also highly involved with the lead hip [45]. Therefore, it is suggested that the increased hip flexibility resulting from the intervention on the lead hip in the present study may have reduced the contribution of lumbar rotation, as the lead hip contributed more to the rotational motion when down swinging a club. The results of this study suggest that the combination of FR and hip stretching to the lead side may be insufficient to significantly reduce the lumbar spine Lrot angle when compared to practicing the golf swing alone. However, the moderate negative correlation between the increase in lead hip IR angle and the decrease in lumbar spine Lrot angle suggests that improving lead hip IR angle may decrease in lumbar Lrot angle during the golf swing.

There are several limitations to this study. Firstly, only immediate effects were examined. A previous study reported that the duration of the effects of FR and stretching was less than 10 min [46]. Hence, the duration of the effect needs to be examined. In addition, we examined only maximal rotation during the golf swing; changes in motion during the entire swing motion are not clear. It is necessary to examine the effects of the FR + DS group motion on the lead hip joint and lumbar spine at different times, such as at ball strike and follow-through. Finally, this study is an in vivo measurement using a three-dimensional motion analyzer and may not be directly comparable to actual joint angles.

## 5. Conclusions

Amateur golfers were studied for hip ROM immediately after FR and stretching to the lead hip with consequent effects on hip and lumbar spine movement during the golf swing. Compared to the subjects in a group who only practiced their golf swing, the group receiving FR + DS showed a significant increase in lead hip IR ROM and hip IR angle during the golf swing. Furthermore, for the group receiving FR + DS, the amount of change in lead hip IR angle and lower lumbar spine Lrot angle were negatively correlated, suggesting that improved lead hip IR angle may influence lumbar spine movement during the golf swing. Thus, the application of FR + DS has immediate effects on improving hip IR ROM and may increase hip IR angle during the golf swing. Limited hip IR ROM while playing golf increases lumbar rotation, which is a risk for LBP. Golfers with lower back injuries have limited lead hip IR ROM, suggesting that FR + DS may be a suitable exercise to reduce rotational stress on the lower back. We suggest that this exercise could be considered as a method for preventing LBP in golfers who have limited lead hip IR ROM.

## Figures and Tables

**Figure 1 healthcare-11-02001-f001:**
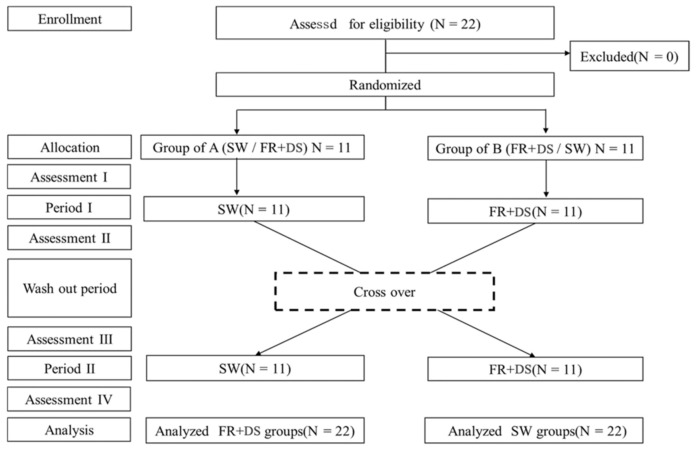
Flowchart of the randomized crossover design. It details patient flow throughout the study. This study had 0 dropouts. Each assessment was conducted before and after each intervention. The crossover period was at least 1 week (mean ± SD: Group of A, 8.1 ± 2.43 days; Group of B, 9.6 ± 3.91 days). SW: Swing practice using a golf club, swinging and hitting a ball. FR + DS: Exercise combined with foam rolling.

**Figure 2 healthcare-11-02001-f002:**
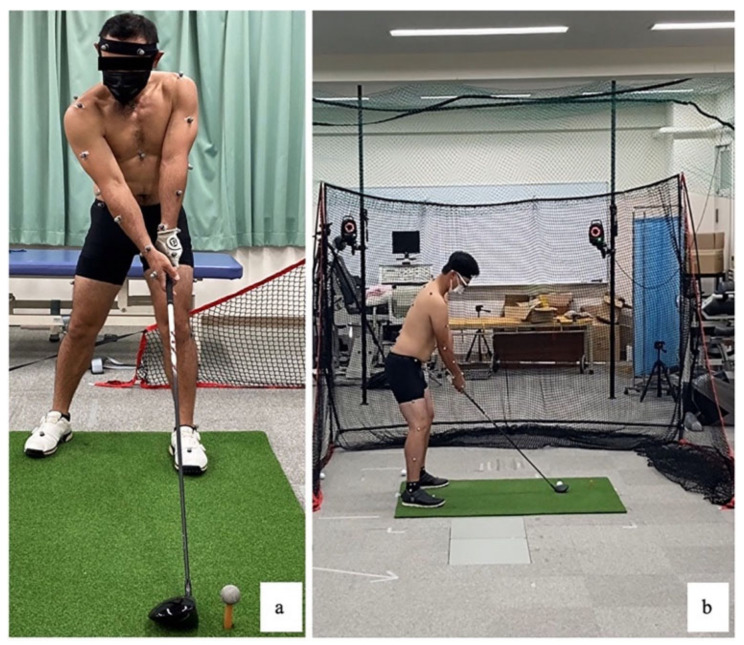
Golf swing measurement. (**a**) Frontal plane. (**b**) Sagittal plane.

**Figure 3 healthcare-11-02001-f003:**
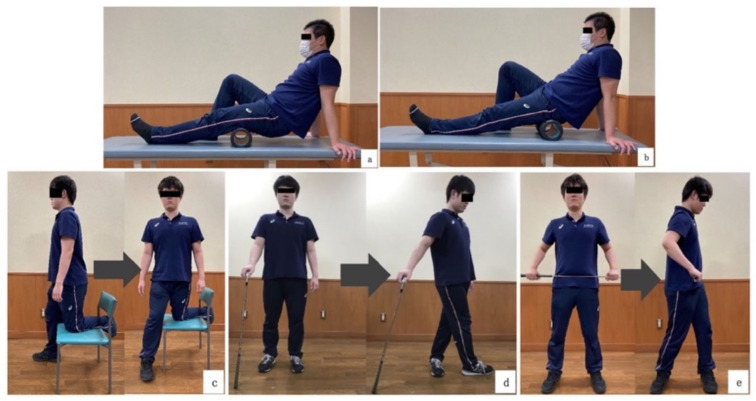
Foam rolling (FR) was performed on the Lead side, distal and proximal sites, for 1 min per site, for a total of 4 min: (**a**) Posterior thigh. (**b**) Buttock. Three hip dynamic stretching exercises (DS): Performed in the order of (**c**–**e**).

**Figure 4 healthcare-11-02001-f004:**
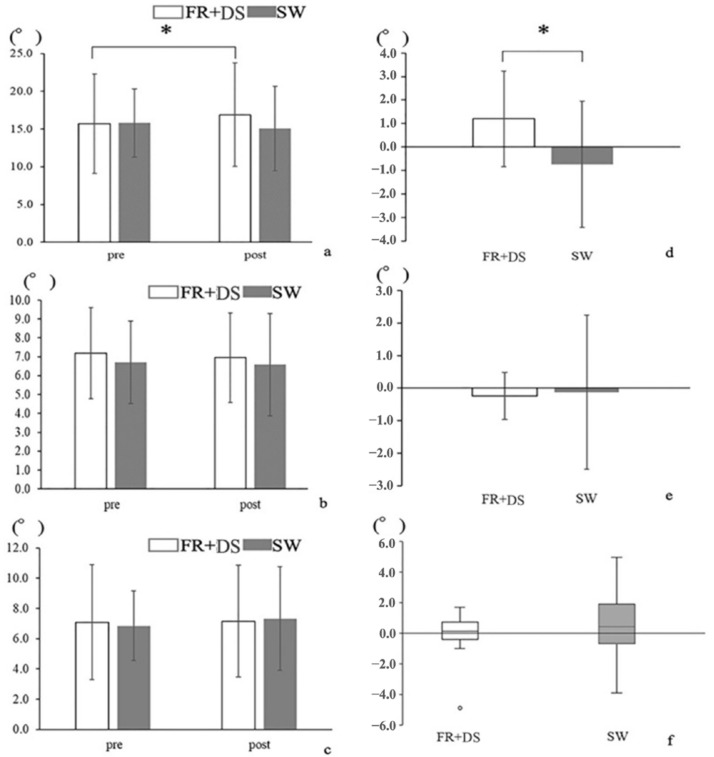
(**a**–**c**): Lead hip and lumbar angle during the golf swing in both groups. (**a**) Lead hip IR angle. (**b**) Upper lumbar Lrot angle. (**c**) Lower lumber Lrot angle. (**d**–**f**): Comparison between groups FR + DS and SW in the amount of change in lead hip and lumbar angle before and after interventions. (**d**) Lead hip IR angle. (**e**) Upper lumbar Lrot angle. (**f**) Lower lumber Lrot angle. The error bars represent SD; however, F is only the lower and upper quartiles. * *p* < 0.05.

**Figure 5 healthcare-11-02001-f005:**
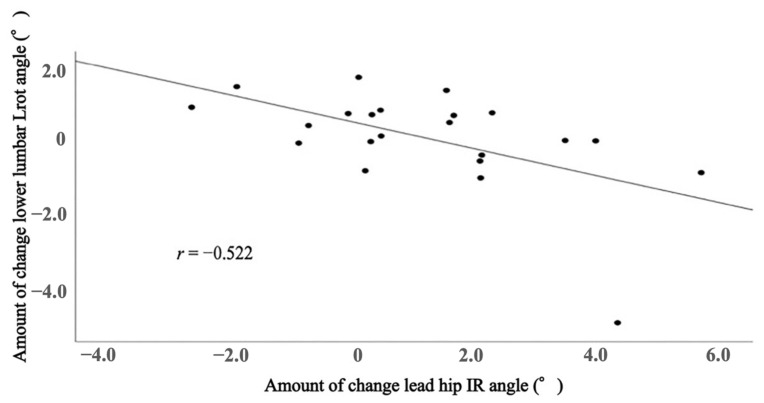
Correlation between lead side hip IR angle and lower lumbar Lrot angle. Spearman’s rank correlation coefficient was used.

**Table 1 healthcare-11-02001-t001:** FR + DS and SW Groups in Pre-exercise of Clinical Test as Secondary Outcomes.

	FR + DS Pre	SW Pre	*p*-Value	95% CI		FR + DS Post	SW Post	*p*-Value	95% CI	
IR ROM										
Lead hip passive	28.7 ± 9.02	28.2 ± 9.04	1.000	−0.38	1.23	35.4 ± 9.74	29.5 ± 7.90	<0.000 *	4.62	7.25
Trail hip passive	26.8 ± 7.66	27.2 ± 8.08	1.000	−1.54	0.74	28.2 ± 7.49	27.6 ± 7.58	0.384	−0.11	1.29
Lead hip active	27.4 ± 8.10	27.7 ± 8.31	1.000	−1.25	0.54	34.4 ± 8.93	28.6 ± 7.35	<0.000 *	4.30	7.18
Trail hip active	26.8 ± 7.57	26.6 ± 7.51	1.000	−1.32	1.88	28.2 ± 7.15	27.1 ± 7.34	0.276	−0.09	2.17
ER ROM										
Lead hip passive	27.8 ± 5.68	27.9 ± 4.95	1.000	−1.42	1.30	27.3 ± 5.40	27.5 ± 4.05	1.000	−1.62	1.10
Trail hip passive	29.1 ± 5.95	28.6 ± 5.26	1.000	−0.62	1.53	28.9 ± 5.12	28.9 ± 5.18	1.000	−1.79	1.68
Lead hip active	26.8 ± 6.33	28.2 ± 4.94	0.068	−2.35	−0.26	27.3 ± 5.12	28.3 ± 4.72	0.280	−2.09	0.09
Trail hip active	28.5 ± 5.78	28.8 ± 4.79	1.000	−1.41	0.95	29.0 ± 4.83	29.3 ± 5.27	1.000	−1.76	1.10

FR + DS: combination foam roller and stretching exercise. SW: swing practice. IR ROM: hip internal rotation range of motion. ER ROM: hip external rotation range of motion. The *p*-value was corrected using Bonferroni correction after performing a paired *t*-test. * The values indicate statistical significance of *p* < 0.05. Mean SD and 95% CI for normally distributed data.

## Data Availability

The datasets generated and/or analyzed during the current study are available in the Mendeley Data at https://data.mendeley.com/datasets/27ywz2kj4v/1 (accessed on 7 March 2023).

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
