# Peer review of "Immediate Effects of Foam Roller and Stretching to the Lead Hip on Golfers Swing: A Randomized Crossover Trial"

_healthcare, 2023, doi:10.3390/healthcare11142001_

Round 1

Reviewer 1 Report

The aim of the present study was to investigate the effects of combined methods of flexibility training (myofascial foam roller and stretching) on the leading hip and lumbar rotation movement during the golf swing. The results of the study showed a significantly greater leading hip internal rotation (IR) angle and range of motion (ROM) during the golf swing after the flexibility exercises. Further, a moderate negative correlation was found between leading hip IR angle and lower lumbar spine rotation movement. The authors concluded that the use of FR combined with stretching is suitable for increasing the leading hip IR angle and ROM and may decrease lumbar spine rotation movement during the golf swing.

The manuscript and the topic are interesting. Below, I provide some comments and suggestions for helping authors to improve the quality of their manuscript.

GENERAL COMMENTS:

Abstract:

Line 26: Please change “lumber” to “lumbar”.

Keywords: Please avoid to include keywords already presented in the title.

Introduction:

Line 35: Please include “being” at the end of the sentence.

Lines 83-84: Please rewrite. This sentence seems to be incomplete.

Materials and methods:

Line 125: Please correct “at least”.

Line 183: Change “…past studies…” to “…previous studies…”.

Discussion:

Lines 270-272: Please rewrite. Difficult to follow.

Lines 310-312: Please rewrite for a better reader’s comprehension.

Conclusions:
Line 339: Please add: “…during golf swing…”.

SPECIFIC COMMENTS:

Introduction section:

Line 59: Regarding exercise as a method for preventing and reducing LBP in golfers, please provide more information following F.I.T.T.-V.P. parameters (i.e., frequency, intensity, time, type, volume, progression).

Lines 68-71: Flexibility training methods, such as stretching, may not be necessarily included in the warm-up of a training session. It would depend on the type of the subsequent exercise, since in some cases it could impair exercise performance (e.g., strength levels are reduced after stretching exercises). I suggest authors to introduce this idea.

Materials and methods:

Line 98: How long did it take the intervention in each period (I and II)? Did participants undertake any familiarization with FR and stretching exercises? It would be interesting to report more information regarding participants’ characteristics (e.g., years of experience, training frequency, golf performance (personal best score for 18 holes, etc)).

Lines 193-196: You stated that participants performed stretching exercises and then you report that they carried out 2 sets of 15 repetitions of each exercise without holding position. Stretching is traditionally considered a static, passive method of flexibility training which implies holding a position for a specific time. Since your participants performed dynamic controlled exercises, I suggest authors to use the term of “dynamic stretching” for a better reader’s comprehension.

Results:

Please report the interaction effects.

Discussion:

I encourage authors to exhaustively revise this section, since there are some sentences that must be rewritten for a better reader's comprehension.

Please let a native English speaker to review your manuscript.

Reviewer 2 Report

Purpose: Investigate the effects of foam rolling to the leading hip muscles combined with stretching on leading hip and lumbar rotation movemente during the golf swing.

Hypothesis: Not presented

Introduction

Lines 49-50: You cite "previous studies" but cited a single reference (#Ref 5). Please perform the concordance between the plural and the number of citations

4th Paragraph: Disproportionate, develop only 4 lines. I recommend developing this paragraph a little further. Create a connection with the paragraph below.

Lines 72: Please change "myofascial release" ot "massage". Current literature doesn't support the nomenclature of "myofascial release". Check throughout the text.

Line 74: Same comments.

Lines 72-76: Long text without references

Introduction well developed but scattered. I recommend grouping these sections into 4-5 paragraphs.

Materials and Methods

Lines 102-103: It's unclear the need to perform a washout.

Figure 1: You indicate a Cross-Over in interventions, which refutes a clinical trials. Please clarify this point.

Line 211: Please indicate the tests and values of the normality tests.

Discussion

I missed discussing with papers that compared FR with stretching techniques, extrapolating the results to your object of study. I recommend inserting this type of literature.

In addition, it is necessary to present the biomechanical and/or physiological mechanisms that explain their findings.
